# “Mmm, Smells like Coffee!”: How a Brief Odor Identification Test Could Help to Identify People with Mild Cognitive Impairment and Dementia

**DOI:** 10.3390/brainsci13071052

**Published:** 2023-07-10

**Authors:** Wolfgang Trapp, Andreas Heid, Susanne Röder, Franziska Wimmer, Göran Hajak

**Affiliations:** 1Department of Psychiatry, Sozialstiftung Bamberg, St.-Getreu-Straße 18, 96049 Bamberg, Germanyfranziska.wimmer@sozialstiftung-bamberg.de (F.W.);; 2Department of Physiological Psychology, Otto-Friedrich University Bamberg, Markusplatz 3, 96045 Bamberg, Germany

**Keywords:** dementia, mild cognitive impairment, Mini-Mental State Examination, odor identification, Sniffin’ Sticks olfactory identification test, smell

## Abstract

(1) Background: Dementia and mild cognitive impairment (MCI) are still underdiagnosed in the general population. Impaired odor identification has been identified as an early marker of MCI and dementia. We aimed to compare the additional diagnostic value of two odor identification tests to a cognitive screening test in detecting MCI or dementia. (2) Methods: The Sniffin’ Sticks odor identification test (SS-OIT), a brief odor identification test (B-OIT) requiring the identification of coffee scent, and the Mini-Mental State Exam (MMSE) were administered to a consecutive series of 174 patients (93 with dementia, 42 with mild cognitive impairment, and 39 without cognitive impairment) referred for neuropsychological testing. (3) Results: Both participants with dementia and with MCI exhibited impairments in odor identification. The SS-OIT and the B-OIT were substantially correlated. Complementing MMSE scores with the SS-OIT or the B-OIT similarly improved the diagnostic accuracy of individuals with dementia and MCI. (4) Conclusions: People with suspected dementia or MCI may already benefit from brief odor identification tests. Although these tests require little additional time, they can notably increase sensitivity for dementia or MCI.

## 1. Introduction

Dementia is defined by the World Health Organization’s International Classification of Diseases (ICD-11) as a significant decline from an individual’s previous level of functioning in two or more cognitive domains not attributable to normal aging that severely limits daily living activities [1]. At the moment, about 55 million people worldwide [2] are affected. This number is predicted to rise to more than 150 million people worldwide by 2050 [3].

Although an early diagnosis of dementia is essential, less than half of the people in the general population with dementia have received a formal diagnosis [4]. However, an early diagnostic would be crucial for several reasons. First, interventions to slow down the progression of cognitive deficits could be initiated. Second, care plans could be implemented while patients still have the legal capacity. Third, institutionalization might be postponed. Furthermore, all these interventions have been proven to enhance the quality of life and delay admission to institutional care [5,6,7].

The conscious perception of an odor arises from the interaction of odor molecules with olfactory receptor neurons in the mucosa of the olfactory epithelium that covers the nasal cavity and the surface of the superior and part of the middle turbinate bones. The axons of the olfactory receptor neurons project to the olfactory bulb. In turn, the axons of the neurons from the olfactory bulb synapse with neurons of the amygdala, the piriform, and the entorhinal cortex, which then project to the orbitofrontal cortex and the hippocampus.

This olfactory system shares a common neurological basis with degenerative regions in dementia, especially with Alzheimer’s dementia (AD), as the connections of the bulbus olfactorius with the entorhinal cortex, the hippocampus, the orbitofrontal cortex, and the amygdala aid in remembering and identification of scents [8,9,10]. Empirical research has shown that amyloid beta levels correlate with olfactory dysfunction in dementia [11,12]; furthermore, in both dementia and mild cognitive impairment (MCI), often conceived as a transitional state between average age-related cognitive decline and dementia [13,14], olfactory function, especially odor identification, is more impaired than would be expected with normal aging [15,16,17]. Moreover, evidence from longitudinal studies suggests that odor identification predicts the progression of unimpaired persons to dementia or MCI and those with MCI to dementia [18,19,20].

Several commercial, reliable, well-validated odor identification (OI) tests, such as the University of Pennsylvania smell identification test or the Sniffin’ Sticks odor identification test, are available [21,22]. In combination with cognitive screenings, these tests seem to increase the diagnostic accuracy of individuals with AD and MCI [23,24]. Furthermore, preliminary results show that even very brief odor identification tests requiring the identification of the smell of peanut butter or coffee powder could contribute to diagnostic accuracy [25,26].

Given the cumulative evidence summarized above, we propose that even a very brief odor identification (B-OIT) test requiring the identification of only one odor in combination with cognitive screening tests might reach the diagnostic benefit of standardized, more extensive OI tests.

In this paper, therefore, we tested the hypotheses that (1) individuals with AD and MCI have lower odor identification scores than healthy older adults in both a standardized test of OI and a brief odor identification screening test (BOIST), (2) the brief screening test and the standardized OIT improve diagnostic classification using the MMSE, and (3) adding the OIT to the BOIST results in no significant increase in diagnostic quality.

## 2. Materials and Methods

### 2.1. Participants

A consecutive series of patients who were referred for neuropsychological testing on a routine basis, because of a suspected cognitive decline or due to their wish, were recruited from the departments of geriatric internal medicine and geriatric psychiatry of a general hospital in Bamberg, Germany. All of them underwent routine laboratory screening, including thyroid function parameters, lues serology, B12 and folic acid levels, a cranial computer tomography (CT) or magnetic resonance imaging (MRI) scan, EEG, ECG, and a thorough neuropsychological, psychiatric, neurological, and physical examination to secure a dementia diagnosis. 

A senior psychiatrist saw all patients. The decision as to whether the examined patient had dementia or MCI was made at a multidisciplinary meeting using ICD-10 criteria for the diagnosis of dementia and additional established criteria [27,28,29,30,31,32,33] for the diagnosis of mild cognitive impairment (MCI) and dementia subtypes. Participants diagnosed with moderate or severe dementia and patients exhibiting significant depressive symptoms were excluded.

### 2.2. Screening Tests and Symptom Measure

Mini-Mental State Examination (MMSE, [34]) as part of a more comprehensive neurological test battery, which included the German version of the Consortium to Establish a Rationale in Alzheimer’s Disease diagnostic neuropsychological battery (CERAD-Plus [35]), the Bamberg Dementia Screening Test (BDST [36]), and the German version of the Frontal Assessment Battery (FAB-D [37]), was administered to all participants. 

The Sniffin’ Sticks odor identification test (SS-OIT, [22]) as a standardized test and a brief odor identification screening test (B-OIT), including the correct identification of coffee scent, were used to test OI [26]. The SS-OIT requires subjects to correctly identify 16 odors administered via pen-like odor dispensers from four given choices. SS-OIT scores can range from 0 to 16, and its administration takes 5–8 min. According to the manual, scores ≤11 indicate impaired odor identification. In the B-OIT, participants must identify the scent of coffee correctly. In contrast to the procedure used by Streit and colleagues [25], the odor stimulus is presented in several steps: First, participants have to close their eyes and identify the scent of coffee powder held 5–10 cm under their nostrils. If their answer is correct, three points are awarded. Otherwise, the four response alternatives “cigarette”, “coffee”, “wine”, and “candle smoke” (corresponding to the choices in the similar item of the SS-OIT) are given. If participants choose the correct alternative, two points are awarded. If the answer is incorrect, subjects may open their eyes and judge again which substance they smell. If coffee is identified correctly, one point is awarded; otherwise, no points are granted. Thus, the score reachable can range from 0 to 3.

Furthermore, all patients completed the German short version of the Geriatric Depression Scale (GDS) [38], a brief screening instrument for depressive symptoms in older adults. Participants with GDS scores above 5, indicating possible depression, were excluded.

### 2.3. Statistical Analyses

Univariate analyses of variance to compare age, GDS scores, and years of education in the three diagnostic groups (CNT, MCI, and mild dementia) were performed. Likewise, univariate analyses of variance using orthogonal contrasts (CNT vs. DEM and MCI; DEM vs. MCI) were conducted to compare the MMSE, SS-OIT, and B-OIT scores. As no significant differences between participants with MCI and dementia concerning the SS-OIT and B-OIT scores were found, the two subsamples were combined.

In order to gain information about the B-OIT’s concurrent and discriminant validity, Pearson correlation coefficients between SS-OIT and B-OIT and between B-OIT and all 16 items of the SS-OIT were computed. In addition, the MMSE, FAB-D, and the CERAD-Plus subtests for verbal learning, verbal recall, Trail Making Test A, and Trail Making Test B were correlated with the SS-OIT and the B-OIT scores.

In order to compare the diagnostic performance of the MMSE, the SS-OIT, and the B-OIT, three stepwise logistic regression analyses were conducted using the diagnostic group (CNT vs. MCI/mild DEM) as the dependent variable and (a) the MMSE and the SS-OIT as predictors, (b) the MMSE and the B-OIT as predictors, or (c) the MMSE, the SS-OIT, and the B-OIT as predictors.

Sensitivity (percentage of participants with cognitive impairments detected by the test) and specificity scores (percentage of participants with no cognitive impairments correctly classified as unimpaired) were separately computed for the MMSE, the SS-OIT, and the B-OIT.

In order to assess the practical benefit of a combined diagnostic use of cognitive and OI screenings, sensitivity and specificity were also calculated in the case of at least one positive result in one of the two tests. As depicted in Figure 1, the use of an OI test in addition to a cognitive screening test should increase sensitivity at the expense of reducing specificity. OI screening should detect some subjects with possible MCI or DEM not correctly identified by the cognitive screening. On the other hand, there is an increased risk that unimpaired persons will be incorrectly classified as impaired in one of the two methods. For this reason, sensitivity and specificity were calculated for different diagnostic scenarios using cutoff scores of ≤26, ≤27, and ≤28 for the MMSE, of ≤11 for the SS-OIT, and of ≤2 and ≤1 for the B-OIT.

## 3. Results

### 3.1. Sample Characteristics

In total, 174 participants, 39 (15 females) with MCI, 93 (61 females) with mild dementia (DEM), and 42 without cognitive impairment (20 females), were included. The latter group was, thus, included as a clinical control sample (CNT).

No significant differences were found in the three groups (CNT, MCI, and DEM) concerning age, GDS scores, and years of education (F_(2,171)_ = 2.922, *p* = 0.057); see Table 1 for more detailed information about the sample.

However, the three groups differed significantly in their MMSE, SS-OIT, and B-OIT test scores. Orthogonal contrasts were significant for the comparison between CNT and MCI/DEM for all measures (t-values between 2.79 and 3.74, *p* between 0.006 and <0.0005) and for the comparison between MCI and DEM for the MMSE (t = 5.67, *p* < 0.0005) but not for the SS-OIT (t = 0.93, *p* < 0.353) and the B-OIT (t = 1.84, *p* < 0.067).

### 3.2. Validity of the B-OIT

The correlation between the SS-OIT and the B-OIT was 0.54 (*p* < 0.0005), and the B-OIT correlated significantly with 13 of the 16 SS-OIT items (r between 0.19 and 0.44, *p* between 0.012 and <0.0005), with the highest correlation resulting for SS-OIT item 10 (“coffee scent”). This result indicates a good concurrent validity of the B-OIT. In contrast, the two odor identification tests had only low correlations with different cognitive tests (see Table 2). Nevertheless, significant correlations resulted for both OI-tests with the MMSE, the verbal learning test, and the TMT B, as well as for the SS-OIT and the verbal recall subtest of the CERAD-Plus.

### 3.3. Diagnostic Performance of the MMST, the SS-OIT, and the B-OIT

The stepwise logistic regression for the MMSE and the SS-OIT predicting the diagnostic group (CNT vs. MCI/DEM) resulted in a model including both predictors (−2 log-likelihood = 157.099; Nagelkerke’s R^2^ = 0.274; see Table 3).

The same logistic regression for the MMSE and the B-OIT also resulted in a model including both predictors (−2 log-likelihood = 153.816; Nagelkerke’s R^2^ = 0.297; see Table 3). 

The inclusion of all three predictors yielded the first model described above, only containing the MMSE and the B-OIT but not the SS-OIT (*χ*^2^_(df = 1)_ = change in −2 log-likelihood for inclusion of the SS-OIT = 0.465, *p* = 0.495). Thus, including the SS-OIT additionally to the MMSE and the B-OIT provides no additional benefit.

Sensitivity and specificity values for the SS-OIT for the established cutoff score of ≤11 alone were 65.9% and 57.1%, respectively. The corresponding sensitivity and specificity values for the B-OIT were 66.7% and 54.8% for a cutoff score of ≤2 and 39.4% and 90.5% for a cutoff score of ≤1.

Figure 2 shows the sensitivity and specificity values for different scenarios for the MMSE used alone and combined with either the SS-OIT or the B-OIT. As can be seen, the sensitivity increased with an odor identification test in addition to the MMSE, regardless of which test and which cutoff scores are used. Conversely, there was a decrease in specificity when an odor identification test was also included in the diagnostic decision regarding whether cognitive impairment is present. However, this decrease was significantly less pronounced for the B-OIT and a cutoff of ≤1 than for the B-OIT and a cutoff of ≤2 or the SS-OIT.

## 4. Discussion

In this study, the additional benefit of odor identification tests in diagnosing cognitive impairment was investigated in a clinical sample of patients who were referred for the clarification of dementia- or MCI-related cognitive decline. In addition to a standardized and validated procedure for odor identification, it was examined whether a very brief test, in which only one odor had to be correctly identified, could also be of diagnostic value. Both OI tests correlated substantially but showed only low correlations with several cognitive measures. These results indicate that OI and cognitive tests each cover unique variance.

Our data confirm previous studies indicating impaired odor identification in dementia and MCI [15,16,17].

Although this study focused on older adults with cognitive impairments and presumably previously intact olfactory function, it should be pointed out that similar dysfunctions can be demonstrated very early. In early infancy, hypoxic/ischemic encephalopathy at birth, some metabolic encephalopathies, and specific congenital malformations of the brain in which the olfactory bulbs are absent or hypoplastic (holoprosencephaly; septo-optic-pituitary dysplasia) can lead to impairment or loss of odor perception [39]. Later on, certain viral infections, including COVID-19 [40] or influenza [41] and cytomegalovirus [42] can affect the respiratory system. Moreover, odor identification performance generally decreases in old age, independent of the development of cognitive impairments due to loss or dysfunction of the sensitive olfactory receptors of the terminal dendrites of the olfactory nerve [43]. 

Thus, it is no surprise that, when analyzed individually, both odor identification tests showed unsatisfactory values for sensitivity and specificity. 

However, for the present sample, it was shown that combining either of the two odor identification tests with the MMSE resulted in an improvement in sensitivity. This increase in sensitivity was achieved at the expense of specificity, which, in our sample, was reduced to unsatisfactory values of partly less than 50% in some combinations. However, this reduction can be accepted if the odor identification test is administered as a component of a more comprehensive neuropsychological test battery. In this case, the additional time required is limited in comparison to the total duration of the examination, and other measures of the test battery might compensate for the lack of specificity.

On the basis of the data presented in this manuscript, if only a brief screening can be performed before possibly referring to a specialized memory clinic for further diagnostics, administering the B-OIT with a cutoff of ≤1 can be considered. In combination with the MMSE at a cutoff of ≤27, for example, the B-OIT achieved sensitivity and specificity values of 74.2% and 71.4% (compared to 60.0% and 81.0% for the MMSE alone). In addition, the B-OIT also correlated significantly with the SS-OIT and, although significantly shorter, showed just as high prognostic validity.

Unfortunately, our results are difficult to compare with other studies which also used brief tests for odor identification because different study designs were used. In the study by Streit and colleagues [25], only patients with normal values in the MMSE (≥27 points) and the clock drawing test were included, and there was no information on the combination with one of the two cognitive screening methods given. Although the stimulus material used was identical (instant coffee powder), the test was given only in a binary form (odor recognized vs. odor not recognized). These differences could explain the lower sensitivity values reported by the authors. Another study by Stamps and colleagues [26] used peanut butter and measured the distance from the left or right nostril from which a scent could be perceived. Although the authors also noted that participants with dementia, MCI, or no cognitive impairment were asked to identify the scent, they did not provide further data on the frequency of correct identification of the scent by the participants.

Lastly, the study of Quarmley and colleagues [23] that also used the SS-OIT reported much higher sensitivity and specificity values for the combination of the SS-OIT with the Montreal Cognitive Assessment (MoCA [44]). However, the authors used a two-step procedure. The diagnostic quality was determined for the MoCa alone in the first step. In a second step, those participants misclassified by the MoCa were reclassified using the SS-OIT. This procedure trivially results in a higher number of correctly classified individuals; however, in the diagnostic process, it is unknown which participants were misclassified by the cognitive screening.

An important limitation of the presented work is that the two odor tests were compared against the MMSE, which is unsatisfactory, especially in discriminating MCI from cognitively nonimpaired persons [45]. Therefore, future studies should include data from more sensitive cognitive tests.

The data presented in this paper were drawn from a clinical sample of patients referred for neuropsychological testing. Although this might be a valid setting in many cases (e.g., in a geriatric ward of a general hospital, a quick assessment might be beneficial), this led to a high proportion of participants with the target condition (dementia or MCI). Our sample size was small; thus, the results should be cross-validated, desirably in a population-based sample.

## Figures and Tables

**Figure 1 brainsci-13-01052-f001:**
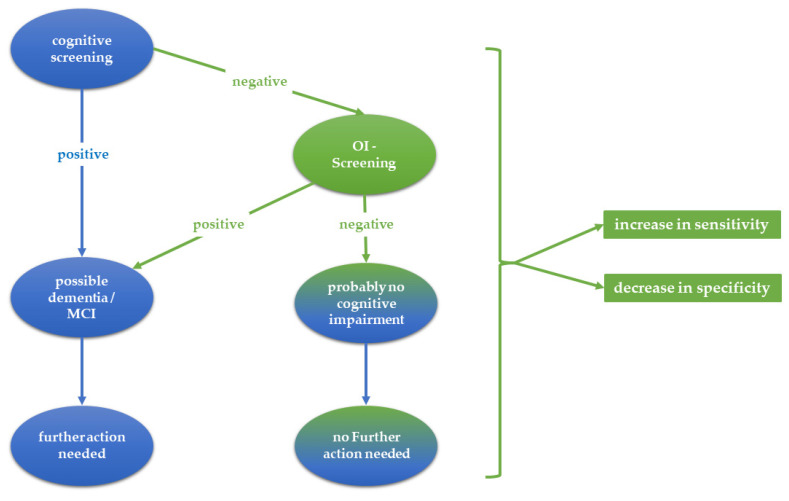
Scenario for performing an odor identification test in addition to a cognitive screening test.

**Figure 2 brainsci-13-01052-f002:**
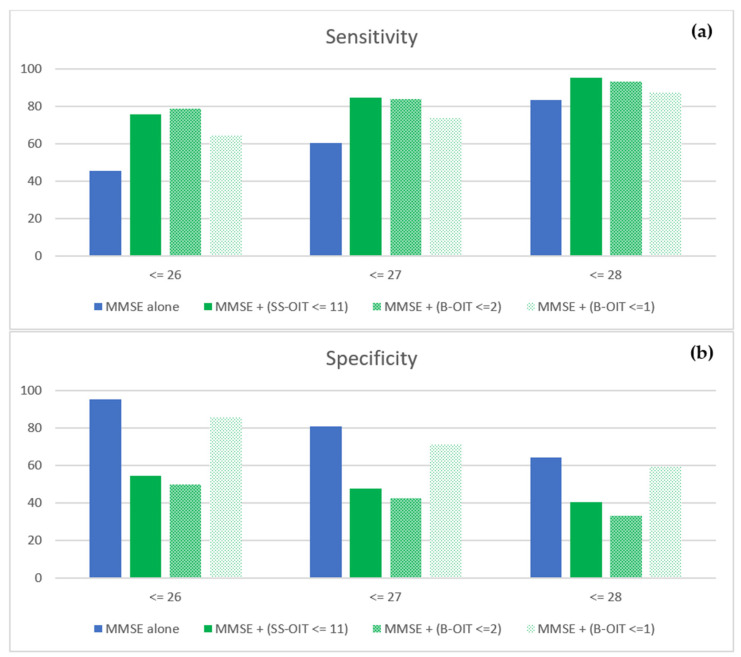
Sensitivity (**a**) and specificity (**b**) values for different combinations of the MMSE, the SS-OIT, and the B-OIT using diverse cutoff scores in the scenario depicted in Figure 1.

**Table 1 brainsci-13-01052-t001:** Sample characteristics. SD: standard deviation, CNT: clinical control sample, DEM: mild dementia, MCI: mild cognitive impairment, GDS: Geriatric Depression Scale, MMSE: Mini-Mental Status Examination, SS-OIT: Sniffin’ Sticks odor identification test, B-OIT: brief odor identification test.

	CNT (*n* = 42)	MCI (*n* = 39)	DEM (*n* = 93)	Analysis of Variance	Orthogonal Contrasts	
					CNT vs. DEM/MCI	DEM vs. MCI
	Mean (SD) [range]	Mean (SD) [range]	Mean (SD) [range]	F_(2,171)_ (p)	T(191) (p)	T(191) (p)
Age	69.57 (9.06) [57–87]	70.95 (10.54) [52–86]	72.56 (8.41) [50–87]	1.652 (0.195)	1.33 (0.187)	0.93 (0.354)
Years of education	13.29 (2.66) [11–17]	13.28 (2.62) [11–17]	12.39 (2.20) [11–17]	2.922 (0.057)	1.02 (0.310)	1.92 (0.056)
GDS	4.52 (4.13) [0–5]	4.56 (4.29) [0–5]	4.38 (3.81) [0–5]	0.034 (0.967)	0.07 (0.947)	0.23
MMSE	28.40 (2.09) [26–30]	28.08 (3.18) [24–30]	25.52 (2.77) [18–30]	28.943 (<0.0005)	3.74 (<0.0005)	5.67 (<0.0005)
SS-OIT	9.76 (0.62) [2–15]	8.81 (1.62) [5–15]	8.78 (1.61) [1–15]	5.120 (0.007)	2.79 (0.006)	0.93 (0.353)
B-OIT	7.04 (0.86) [0–3]	5.19 (1.75) [0–3]	4.84 (1.81) [0–3]	8.147 (>0.0005)	3.11 (0.002)	1.84 (0.067)

**Table 2 brainsci-13-01052-t002:** Correlations of the two OI tests with cognitive measures. r: Pearson correlation coefficient, MMSE: Mini-Mental Status Examination, SS-OIT: Sniffin’ Sticks odor identification test, B-OIT: brief odor identification test, CERAD-Plus: German version of the Consortium to Establish a Rationale in Alzheimer’s Disease diagnostic neuropsychological battery, FAB-D: German version of the Frontal Assessment Battery, TMT: Trail Making Test.

		B-OIT		SS-OIT	
		*r*	*p*	*r*	*p*
MMSE		0.19	0.012	0.22	0.003
CERAD-Plus	Verbal learning	0.21	0.006	0.25	0.001
	Verbal recall	0.15	0.044	0.10	0.173
	TMT A	−0.12	0.118	−0.09	0.258
	TMT B	−0.19	0.012	−0.18	0.018
FAB-D		0.11	0.173	0.14	0.068

**Table 3 brainsci-13-01052-t003:** Results of logistic regression analyses to predict cognitive impairment. MMSE: Mini-Mental Status Examination, SS-OIT: Sniffin’ Sticks odor identification test, B-OIT: brief odor identification test, OR: odds ratio, SE: standard error, β: standardized regression coefficient.

		β	SE(β)	*p*	OR
Model for MMSE and SS-OIT	MMSE	0.581	0.143	<0.0005	1.788
	SS-OIT	0.148	0.074	0.047	1.159
Model for MMSE and B-OIT	MMSE	0.569	0.141	<0.0005	1.766
	SS-OIT	0.614	0.237	0.010	1.847

## Data Availability

The dataset used and analyzed during the current study is available from the corresponding author on reasonable request.

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
