# Peer review of "“Mmm, Smells like Coffee!”: How a Brief Odor Identification Test Could Help to Identify People with Mild Cognitive Impairment and Dementia"

_brainsci, 2023, doi:10.3390/brainsci13071052_

Round 1

Reviewer 1 Report

This is an interesting study on the value of a brief odor test in assisting in the diagnosis of MCI and mild dementia. This test (B-OIT) is compared against MMSE and a standardized odor test (OIT).

Abstract

Please compare the text that is included in the manuscript (pdf) and the test shown in the platform (mention of MoCA?)

Methods

A brief description of the neuropsychological battery administered to all patients toward establishing diagnosis of MCI or dementia (line 77) should be included in the methods 

Only the MMSE is mentioned in the abstract and Methods but there is mention of two cognitive screening instruments in the last paragraph of the Introduction.

Results

For clarity Table 1 should include three additional columns with p values for all pairwise contrasts. Value ranges would also be helpful

par 3.2. Concerning discriminant validity, what was the correlation of the two odor tests with cognitive scores (MMSE and, to avoid ceiling effects, 1-2 other neuropsychological tests, for instance of memory and processing speed/attention)?

Perhaps the most crucial limitation of the presented work is that the two odor tests are compared against the MMSE which performs very differently in discriminating MCI from cognitively non-impaired and dementia from cognitively non-impaired persons. For detecting MCI, typically, the MMSE has relatively poor sensitivity among persons with high educational attainment and poor specificity among persons with very low education. This may also be true for persons with relatively mild cognitive impairment who meet criteria for dementia (probably due to evidence of impaired daily function) as in the present sample. Thus lumping together the two impaired groups in comparisons with MMSE is not valid (although the fact that [most?} of the patients in the dementia group may present with relatively mild cognitive deficits). This makes including data from more sensitive cognitive tests (MoCa and/or neuropsychological tests) very important.  

A table with beta values and ORs for each model would be easier to read

Is 11 points a clinically validated cutoff for the OIT (it should be mentioned in the Methods). 

Author Response

  1. Please compare the text that is included in the manuscript (pdf) and the test shown in the platform (mention of MoCA?)
    You are right. Thank you for pointing that out. The abstract in the manuscript text is correct (MMSE instead of MoCA). Unfortunately, it does not seem possible to change the abstract text in the platform afterward.
  2. A brief description of the neuropsychological battery administered to all patients toward establishing diagnosis of MCI or dementia (line 77) should be included in the methods 
    Thank you very much for this hint. We have included this information (CERAD-Plus, BDST, FAB).
  3. Only the MMSE is mentioned in the abstract and Methods but there is mention of two cognitive screening instruments in the last paragraph of the Introduction.
    Oh, thank you for drawing attention to this! That is because we usually also intended to share data for the BDST (developed in our workgroup) and then decided that this would unnecessarily complicate the results section (the results parallel the MMST results with the exception that BDST has better SEN and SPE). We have changed this sentence accordingly.
  4. For clarity Table 1 should include three additional columns with p values for all pairwise contrasts. Value ranges would also be helpful
    As described in the methods section, we used the maximum number of two orthogonal contrasts in our analyses of variance (CNT vs. DEM/MCI and DEM vs. MCI) to avoid compromising the alpha level. Therefore, our updated Table 1 has only two additional columns. Would you like us to calculate Scheffé a posteriori comparisons for all three possible comparisons?
  5. par 3.2. Concerning discriminant validity, what was the correlation of the two odor tests with cognitive scores (MMSE and, to avoid ceiling effects, 1-2 other neuropsychological tests, for instance of memory and processing speed/attention)?
    Thank you, this is an excellent point. We have added a table containing correlations of the two OI-Tests with the MMSE, two verbal learning tests, TMT A/B of the CERAD-Plus, and the FAB-D.
  6. Perhaps the most crucial limitation of the presented work is that the two odor tests are compared against the MMSE which performs very differently in discriminating MCI from cognitively non-impaired and dementia from cognitively non-impaired persons. For detecting MCI, typically, the MMSE has relatively poor sensitivity among persons with high educational attainment and poor specificity among persons with very low education. This may also be true for persons with relatively mild cognitive impairment who meet criteria for dementia (probably due to evidence of impaired daily function) as in the present sample. Thus lumping together the two impaired groups in comparisons with MMSE is not valid (although the fact that [most?} of the patients in the dementia group may present with relatively mild cognitive deficits). This makes including data from more sensitive cognitive tests (MoCa and/or neuropsychological tests) very important. 
    We totally agree with you. However, the problem with our data set is that the more sensitive tests (like the CERAD-Plus or the FAB) were unfortunately used to diagnose mild dementia or MCI, while the MMSE was only used to differentiate between mild and moderate dementia. Problem is that the more sensitive date was used to diagnose MCI or (mild) dementia. At least, we have included your arguments in the discussion section now.
  7. A table with beta values and ORs for each model would be easier to read
    Good idea, thank you. We have included a new table. I have reversed the group codes for the logistic regressions to achieve ORs > 1 for your convenience. Therefore the beta values are now also positive.

  8. Is 11 points a clinically validated cutoff for the OIT (it should be mentioned in the Methods). 
    Exactly, thank you very much. We have mentioned this in the “methods” section now.

Reviewer 2 Report

My confidential comments to the Editor is that this is a well done, original study which is clinically useful and I would recommend MINOR REVISION. I have made a few suggestions to the authors for further improvement. I would be happy to read their revision if you offer them this opportunity and if you would like my further input. This manuscript is quite appropriate for Brain Sciences and requires no major copy-editing of the English. I would be most appreciative of your confirmation that my review has been received. Thank you again for having invited me to participate as a peer-reviewer.

Author Response

  1. The first is the developmental aspect of olfactory function. Reliable and reproducible olfactory reflexes and discrimination can be elicited in the neonate and even in premature infants from 30 weeks gestation, as well as demonstrated in fetuses of the 3rd trimester (Sarnat HB, Flores-Sarnat L. Olfactory development. Part 1. Functional, from fetal perception to adult wine-tasting. J Child Neurol 2017;32:566-578). Impaired olfactory responses occur in early infancy with hypoxic/ischaemic encephalopathy at birth, in some metabolic encephalopathies and in certain congenital malformations of the brain in which the olfactory bulbs are absent or hypoplastic (holoprosencephaly; septo-optic-pituitary dysplasia). Though this present study is focussed upon adults with dementia and presumably previously intact olfactory function, it would be useful to point out to the reader that similar dysfunction can be demonstrated from early ages.
    Excellent point. We have added this information in the discussion section of our manuscript. Thank you for the excellent and very informative reference ?.
  2. The second item that could merit more attention in the Discussion is the loss of olfaction in older children and adults with certain viral infections, most notably in recent years with COVID-19 but also with influenza, cytomegalovirus and other infections that primarily affect the respiratory system.
    This argument can now also be found in the discussion section. Many thanks for your hint.
  3. On page 6, line 217 of the manuscript, the authors state, “…cognitive impairment due to thinning of the olfactory mucosa…” and cite two references to document their statement. The main reason for impaired sense of smell most likely is due to loss or dysfunction of the sensitive olfactory receptors of the terminal dendrites of the olfactory nerve. The olfactory epithelium covers not only the nasal cavity but also the surface of the superior and part of the middle turbinate bones.
    Thank you very much. We have changed the corresponding section of the discussion accordingly.
  4. Whereas this manuscript has a focus of clinical testing of olfactory function, briefly addressing the anatomy of the peripheral olfactory system would not be too unrelated to the main theme.
    Thank you for this valuable hint. A brief description of the olfactory system can now be found in the “introduction” section.

And finally: Thank you very much for the compliments on our manuscript ?.